# Lipid and Polymer-Based Nanoparticle siRNA Delivery Systems for Cancer Therapy

**DOI:** 10.3390/molecules25112692

**Published:** 2020-06-10

**Authors:** Francesco Mainini, Michael R. Eccles

**Affiliations:** Department of Pathology, Dunedin School of Medicine, University of Otago, Dunedin 9054, New Zealand; francesco.mainini@gmail.com

**Keywords:** nanoparticle, intracellular delivery, siRNA, cancer therapy

## Abstract

RNA interference (RNAi) uses small interfering RNAs (siRNAs) to mediate gene-silencing in cells and represents an emerging strategy for cancer therapy. Successful RNAi-mediated gene silencing requires overcoming multiple physiological barriers to achieve efficient delivery of siRNAs into cells in vivo, including into tumor and/or host cells in the tumor micro-environment (TME). Consequently, lipid and polymer-based nanoparticle siRNA delivery systems have been developed to surmount these physiological barriers. In this article, we review the strategies that have been developed to facilitate siRNA survival in the circulatory system, siRNA movement from the blood into tissues and the TME, targeted siRNA delivery to the tumor or specific cell types, cellular uptake, and escape from endosomal degradation. We also discuss the use of various types of lipid and polymer-based carriers for cancer therapy, including a section on anti-tumor nanovaccines enhanced by siRNAs. Finally, we review current and recent clinical trials using NPs loaded with siRNAs for cancer therapy. The siRNA cancer therapeutics field is rapidly evolving, and it is conceivable that precision cancer therapy could, in the relatively near future, benefit from the combined use of cancer therapies, for example immune checkpoint blockade together with gene-targeting siRNAs, personalized for enhancing and fine-tuning a patient’s therapeutic response.

## 1. Introduction

The discovery of RNA interference (RNAi) in 1998 by Fire et al. [1], laid the foundations for the development of new gene-targeting methodologies based on RNA oligonucleotides. More recently, the endogenous RNAi machinery in mammalian cells has been studied intensively, leading to the discovery of molecular mechanisms that allow for precise regulation of gene expression mediated by double-stranded RNA (dsRNA). DsRNAs, introduced into target cells using a delivery vector, are processed by Dicer, an RNAse III family member, which cleaves the dsRNA molecules into 19–23 nucleotide fragments that contain a 5′ phosphorylated end and an unphosphorylated 3′ end, with two unpaired nucleotide overhangs at each end. These small dsRNAs are called small interfering RNAs (siRNAs). The *N*-domain unwinding activity of Argonaute (Ago)-2 unwinds the siRNA duplex into two single strands: the guide and passenger strands. Once unwound, the guide strand is incorporated into the RNA interference specificity complex (RISC), while the passenger strand is degraded. The RISC complex then binds to an endogenous mRNA that is complementary to the guide strand and cleaves the target mRNA through the separate endonuclease activity of Ago-2. These events affect the stability of target mRNAs leading to their degradation [2,3]. In addition to siRNAs, the dsRNA “targeting” sequences loaded into the RISC complex may also be derived from microRNAs (miRNAs), or from short-hairpin RNAs (shRNAs) (Figure 1). MiRNAs are natural dsRNA molecules produced by all cells, which impact the function of many genes by blocking target mRNA translation [4]. These RNA duplexes are produced from a stem-loop structure called the precursor miRNA and are processed into short dsRNAs by Dicer. Due to the short recognition length requirement, an individual miRNA is able to bind to multiple mRNAs, and hence it has the ability to regulate multiple genes due to reduced binding specificity. This also results in decreased efficiency of gene-silencing for any given gene, as compared to siRNAs. On the other hand, shRNAs are engineered in the laboratory as plasmids. RNA molecules with a tight hairpin turn are expressed from the plasmid, which can be used to facilitate long-term silencing of target gene expression via RNAi [5]. Expression of an shRNA in cells may therefore typically be accomplished by intra-cellular delivery (for example, by transfection) of a plasmid containing specific shRNA sequences, able to target mRNA strands after being processed by Dicer. ShRNA plasmids have the additional advantage of being DNA-based, and so are more resistant to degradation than dsRNAs. However, shRNAs require the use of an expression vector, and so additional transcriptional steps are needed prior to the generation of dsRNA.

Small interfering RNA (SiRNA), short-hairpin RNA (shRNA), and microRNA (miRNA) exert their activity in the cytoplasm of target cells, where they are incorporated into the RISC complex. However, in contrast to siRNAs, shRNAs and miRNAs must be previously processed by Dicer. After binding to the complementary mRNA sequence, Ago-2 mediates cleavage, and subsequent mRNA degradation. SiRNAs are exogenous dsRNAs, while miRNAs are derived from endogenous miRNA genes that are transcribed into primary miRNAs. ShRNAs are transcribed from a plasmid delivered to target cells.

By design, RNAi therapeutics can be targeted to facilitate the downregulated expression of specific genes, and RNAi is emerging as a form of treatment for a number of human diseases, including cancer. Multiple critical characteristics of tumor cells can, for example, potentially be targeted by specific RNAi therapies, aimed at reducing tumor burden and chemoresistance [6,7,8]. However, the clinical application of RNAi therapy remains limited. A major reason for this is that siRNA therapeutics must overcome physiological and cellular barriers, hindering access of siRNAs to the cytoplasm of target cells (Figure 2), where they are able to fulfill their regulatory function. In addition, due to the ubiquitous presence of RNAses, siRNAs also need to avoid enzymatic degradation. To enable siRNA stability and efficient delivery to the cytoplasm of target cells, a biocompatible delivery system is necessary, because in the absence of a delivery vehicle, naked siRNAs have relatively poor pharmacokinetic properties [9].

Recent advances in the field of nanotechnology have led to the development of novel delivery systems that are able to deliver siRNAs to cells in the tumor microenvironment (TME) where they can affect both cancer cells and infiltrating immune cells [10]. Nanoparticles (NPs) have been used as carriers for a large variety of drugs including chemotherapies [11], small molecule inhibitors [12], imaging agents [13], and immune-modulatory drugs [14]. SiRNAs can be incorporated into an NP formulation through covalent bonds with the NP components or by electrostatic interactions with the NP surface, due to their strong negative charge. NP-based delivery systems can also be used to deliver shRNAs [15] or miRNAs [16]. However, this review focuses on recent advances in non-viral siRNA delivery systems, including lipid, polymer, and inorganic-based NPs aimed at delivering siRNAs to tumor cells or immune cells for cancer treatment and the induction of anti-tumor immune responses. Moreover, several of the characteristics of these lipid, polymer and inorganic-based NPs, including abilities for the formation of siRNA nanoparticles, and endosomal escape, are characteristics that we believe make these materials ideal candidates for in vivo siRNA delivery. This review discusses primarily the delivery of NPs via the intravenous route.

## 2. Challenges in siRNA Delivery: Physiological and Intracellular Barriers

In vivo delivery of siRNA has many challenges. Firstly, unmodified and unprotected siRNAs are unstable in serum, as they are easily degraded by RNAses [9]. Multiple strategies that involve chemical modifications of the backbone or the bases of oligoribonucleotides have been used to protect siRNAs without impairing their capacity to bind target mRNA [17]. Secondly, siRNAs injected into the bloodstream are very susceptible to removal by renal clearance, which results in a short siRNA half-life in blood [18]. NP-based delivery systems have the ability to protect siRNAs from intravascular degradation and reduce the risk of degradation and/or interaction with non-target molecules. However, NPs need to be designed in ways to avoid a number of physiological barriers (Table 1), which limits their ability to be delivered to target cells. For some delivery systems, the NP-based siRNA delivery systems are not required to reach the TME to be effective anti-cancer treatments. For example, cancer vaccines, which only need to be recognized by patrolling immune cells, can be injected subcutaneously. Lastly, irrespective of the target cell, siRNAs must be delivered to the cytoplasm of cells to fulfill their regulatory function and degrade target mRNA molecules, which necessitates the bypassing of the endosomal-lysosomal pathway.

### 2.1. Physiological Barriers That Limit NP Accumulation in Tumors

Methodologies such as electroporation and sonoporation have been used to induce the transfer of genetic material inside cells by creating transient pores on the cell membranes [19]. However, these delivery techniques have limitations for in vivo delivery, such as lack of tissue penetration, cell and tissue damage induction and frequently very low transfection efficiency. Therefore, more discreet and efficient methods are desirable to achieve siRNA delivery to tumor cells. Systemic delivery via intravenous administration is the strategy of choice when the target sites are not locally confined or not readily accessible, such as in the vast majority of tumors in humans. By complexing siRNAs with synthetic materials, glomerular filtration and renal clearance can be avoided if nanocomplexes have a hydrodynamic diameter (HD) >6 nm [20]. This discovery allowed for the improvement of siRNA pharmacokinetic properties by ensuring the HD is >6 nm in a large variety of formulations. However, complexation of siRNAs into these relatively large-sized NP has drawbacks, as nanosystems are partially retained by the reticuloendothelial system (RES), which is composed of phagocytic cells, such as circulating monocytes and tissue-resident macrophages, and which recognizes NP as foreign objects. Furthermore, in the bloodstream, NPs encounter other sensor cells including leukocytes, platelets, monocytes, and dendritic cells (DC) all of which are capable of removing NPs from the circulation by phagocytosis [21]. To limit their uptake by immune cells, long branched polymers such as polyethylene glycol (PEG) have been used to generate stealth NPs by minimizing nonspecific interactions with phagocytes and other non-target tissues [22]. Other strategies to avoid uptake rely on the incorporation of ‘’do not eat me’’ signals into the structure of NPs [23]. The implementation of these strategies is to reduce the involvement of the immune system, which may lead to thrombogenicity and complement activation, resulting in altered biodistribution of NP and potential toxicity. More recently, a RES-specific blocking system based on a CD47-derived peptide ligand was utilized as a pre-treatment prior to intravenous injection of NP, resulting in a longer half-life and reduced uptake by macrophages [24]. This report strongly suggested that NP clearance could be reduced by altering the recognition of NP by phagocytes.

In order to target cancer cells, NP-based siRNA delivery systems must extravasate and move through the extracellular matrix (ECM) to accumulate in the TME. Based on the enhanced permeability and retention (EPR) effect, NPs ranging in size from 30 to 200 nm passively accumulate in tumors to a greater extent than in normal tissue [25,26]. This effect occurs when newly formed tumor vessels are abnormal in their arrangement and architecture, a characteristic of tumor vasculature. Abnormalities in the tumor vasculature are characterized by wide fenestrations, which allow the extravasation of nanomedicines, leading to relatively effective and selective accumulation in tumors. In addition, NPs exhibit Brownian random walks through the spaces between network structures in the ECM and are influenced by components of the matrix in several ways. For example, they collide with matrix fibers (steric interactions) and as they move near to fibers, their diffusion is slowed by restricted thermal motion of water molecules (hydrodynamic interactions) [27].

Another important interaction to take into consideration is the formation of the so-called “protein corona” on the surface of an NP after contact with serum proteins [28]. This coating can affect NP size [29], shape [30], and charge [30], which can lead to unexpected changes in tumor accumulation. The effects of the ECM and serum proteins on NPs can be partially mitigated by the addition of PEG, which confers a neutral charge to the delivery system, preventing interaction with charged proteins and ECM components [27].

Another strategy used to induce accumulation of NPs in the TME is the incorporation of targeting ligands, which have a high affinity for surface antigens expressed by cancer cells. In fact, the identification of novel surface antigen alterations in cancer cells has allowed the development of active targeting strategies. These strategies take advantage of different receptors or antigens expressed on cancer cells compared to normal tissues and allow for a higher rate of internalization of NPs by cancer cells expressing a specific targeting receptor. A wide variety of targeting molecule has been used for this purpose, such as antibodies [31], peptides [32,33], small molecules [34,35], polysaccharides [36], and aptamers [37]. To reduce off-target effects, siRNAs can be designed to target cancer-specific mRNAs, thus limiting their effect on non-target cells.

### 2.2. Intracellular Barriers and Endosomal Escape Strategies

If the delivery system can enable siRNAs to be taken up efficiently by target cells, then subsequently siRNAs remain trapped inside endosomes in the absence of an endosomal escape strategy, leading to their degradation. Endosomal vesicles are important for the correct delivery of cellular components and nutrients to specific cell compartments. However, once endocytosed, NPs are transported in early endosomal vesicles, and then later fuse with late endosomes, which are characterized by low pH, maintained through the activity of ATPase-dependent proton pumps. Late endosomes then fuse with lysosomes, exposing the endosomal contents to an even more acidic environment (pH 4.5–5), which is able to efficiently degrade nucleic acids, particularly RNA molecules, which is also enhanced by the presence of specific RNAses. Thus, siRNAs that don’t escape endosomes are degraded, rather than being released into the cytoplasm to function as RNAi effectors. Therefore, NPs must encompass an endosomal escape strategy to deliver the attached siRNA cargo outside the endosomal compartment.

Strategies such as the incorporation of fusogenic molecules can be used to enable the fusion of NPs with endosomal membranes, thereby allowing the release of endosomal contents into the cytoplasm [38]. Other endosomolytic agents include peptides, proteins, polymers, or small molecules like chloroquine [39], which can also be included in NP formulations. For instance, pH-sensitive molecules can facilitate proton sponge effects, and subsequently the release of molecules contained inside endosomes [40]. The proton sponge effect induces an influx of chloride ions, which then results in the destruction of an endosome and the subsequent release of the endosomal contents into the cytoplasm. Two amino acids, arginine and lysine, are protonated at physiological and acidic pH, and can lead to the destabilization of cellular membranes. Polymers of these amino acids, which have been covalently linked to NPs have been widely used to induce endosomal release of siRNAs and other nucleic acids [41,42] (Figure 2).

NPs interact with the cellular membrane (1) and are engulfed inside early endosomal vesicles (2). Within the low pH environment of the endosome, pH-sensitive materials in the NP react, allowing for the creation of pores in the membrane (3), and the subsequent release of the siRNA into the cytoplasm (4). The siRNA can then interact with the RISC complex (5), which mediates the degradation of the target mRNA (6).

Other amino acids such as aspartic and glutamic acids have pH-sensitive properties, since they are positively charged at low pH, while they are not protonated at pH 7. Endosomolytic peptides mimic the activity of pathogens, which have evolved to avoid the phagolysosome trap by creating pores in the endosomes, allowing them to escape [43]. Fusogenic lipids such as 1,2-dioleoyl-sn-glycero-3-phosphoethanolamine (DOPE) have also been used in lipid-based particles to improve endosomal escape. DOPE undergoes phase transition from the lamellar phase to the inverted hexagonal phase at low pH, producing destabilization of the endosomal membrane [44]. DOPE can be used in combination with PEG or other polymers. For example, Hatakeyama and colleagues have developed a DOPE-PEG conjugate, which is cleaved by matrix metalloproteinase (MMP), an enzyme that is highly expressed in cancer cells [45]. An additional escape strategy relies on a novel class of light-triggered molecules, used to facilitate endosomal release by photochemical internalization (PCI). The activation of photosensitizers by light generates reactive oxygen species (ROS), which are able to disrupt endosomes [46]. In addition, photo-controlled delivery systems can be utilized in combination with photodynamic therapy (PDT)—a strategy to eradicate tumors photo-dependently by local irradiation with light [47].

Lastly, it was shown that siRNAs have off-target effects by acting on non-targeted cells, such as innate immune cells. In their endosomes, nucleic-acid sensors of the Toll-like receptor (TLR) family are activated by nucleic acids. Specifically, TLR3, TLR7, and TLR8, are able to recognize siRNAs and initiate a molecular cascade that leads to cell activation with the consequent secretion of cytokines [48]. Both the sequence and structure of siRNAs can be adjusted to minimize the activation of innate immune cells [49]. On the other hand, in the TME, the activation of immune cells by siRNAs could be beneficial to further enhance the immune response against cancer cells. Therefore, it is important that siRNA characteristics are tailored carefully, to either avoid or to induce immune activation.

Taking into consideration these different barriers, siRNA therapeutics requires suitable delivery methods that are able to enhance tumor accumulation, protect siRNAs from degradation, and induce endosomal escape, while at the same time minimizing off-target effects. To address these challenges, a wide variety of bio-compatible nanovectors are currently under investigation. The next section summarizes the most recent advances in lipid and polymer-based NP development, and their pre-clinical application in cancer therapy.

## 3. Lipid and Polymer-Based siRNA Carriers for Cancer Therapy

### 3.1. Liposomes

Liposomes are spherical vesicles composed of at least one lipid bilayer with an aqueous core. The liposomal membrane can be positively or negatively charged, depending on the phospholipid composition. SiRNAs can be incorporated into positively charged liposomes by electrostatic interactions forming lipoplexes [50]. Cationic lipids including 1,2-dioleoyl-3-trimethylammonium propane (DOTAP), *N*-[1-(2,3-dioleoyloxy) Propyl]-*N*,*N*,*N*-trimethylammonium chloride (DOTMA) have been used in combination with neutral lipids such as cholesterol (Chol), DOPE, 1,2-dioleyl-sn-glycero-3-phosphocholine (DOPC), and 1,2-dioleoyl-sn-glycero-3-phosphoethanolamine (DSPE) to form lipoplexes. In most cases, the siRNA–cationic lipid complexes have a much-reduced positive charge. However, electronegative or neutral liposomes have superior pharmacokinetic properties and are more bio-compatible than cationic ones. The inclusion of PEG chains dramatically reduces the positive charge on the surface of liposomes. On the other hand, PEG chains limit the uptake of lipoplexes by tumor cells, ultimately lowering their transfection efficiency [51]. Additionally, PEG interferes with endosomal escape, resulting in siRNA degradation. The incorporation of a pH-sensitive molecular bridge between PEG and other components of the liposome can facilitate the endosomal release of siRNAs, increasing silencing efficiency. Shuian-Yin and colleagues have developed polymer-based liposomal complexes (SPLexes) composed of a pH-sensitive folate-PEG, carrying a vascular endothelial growth factor (VEGF)-targeting siRNA. The folate receptor CD44 is highly expressed in a wide variety of tumor types and can be exploited for specific tumor targeting. SPLexes showed an increased uptake by cancer cells expressing the folate receptor and were able to induce 75% downregulation of the target protein, demonstrating excellent translational potential [52]. More recently, PEGylated DC-Chol/DOPE cationic liposomes were successfully used to treat ovarian cancer in a xenograft murine model. This formulation was designed to downregulate kinesin spindle protein (KSP) in ovarian cancer cells (HeyA8-MDR) to reduce paclitaxel (PDX) resistance. HeyA8-MDR tumor-bearing mice treated with liposomes, containing both PDX and the KSP-targeting siRNA (siKSP), showed reduced tumor growth compared to controls. In addition, KSP protein levels were highly downregulated in excised tumors, demonstrating the in vivo activity of siKSP. Biodistribution studies showed high levels of accumulation of liposomes in tumors and long half-life in the blood (16.5 h) [53]. Another similar formulation was developed by Sheng Yu and colleagues, showing synergistic activity of PDX and Polo-like kinase 1 (PLK-1)-targeting siRNA in limiting the progression of breast cancer. These cationic liposomes were also functionalized with a targeting aptamer (AS1411) to further enhance tumor accumulation [54]. In another report, the use of gemcitabine (Gem) in combination with Myeloid cell leukemia 1 (MCL1)-targeting siRNA in loaded liposomes was shown to be effective in treating pancreatic cancer [55]. These experimental examples suggest that the use of siRNAs targeting specific oncogenes over-expressed in cancer cells are able to synergize with chemotherapy, resulting in reduced tumor growth in a wide variety of xenograft tumor models.

Specific antigen-targeting monoclonal antibodies (mAbs) can be coupled to liposomes in order to achieve specific cell targeting. However, the attachment of mAbs by covalent methodologies to the components of the lipidic bilayer is inefficient and requires careful optimization [56,57]. Recently Kedmi and colleagues developed a flexible coupling technique called ASSET (anchored secondary scFv enabling targeting) [58]. ASSET is a membrane-anchored lipoprotein that can be incorporated into lipoplexes, and by interacting with the antibody crystallizable fragment (Fc) domain of immunoglobulins, it enables the functionalization of an NP with potentially any antibody. With this methodology, the antibody variable domain is exposed for ligand binding, in contrast to standard coupling procedures, which can limit the functionality of the attached mABs. One of the formulations proposed was able to improve survival in a mantle cell lymphoma xenograft model. Another report by Guan et al. [7], showed that active tumor targeting was not always necessary to achieve good therapeutic effects. The authors developed liposomes with a cationic core for siRNA loading and an outer layer composed of DSPE-PEG2000 to prolong circulation. These NPs, loaded with PDX and glyceraldehyde-3-phosphate dehydrogenase (GAPDH)-targeting siRNA, were devised to reduce chemoresistance and limit ATP production in tumors. Most tumor cells are characterized by enhanced glycolysis and high levels of GAPDH, which catalyzes the sixth catabolic reaction of glycolysis (conversion of glyceraldehyde 3-phosphate to D-glycerate 1,3-bisphosphate). Intravenous injections of siGAPDH-PDX liposomes led to a reduction in tumor burden in a murine xenograft model (Hela) with good accumulation in the TME, mediated by the EPR effect. Although this effect plays a critical role for NP accumulation in murine xenograft models, experimental evidence for the effectiveness of EPR in human tumors remains contradictory [50].

### 3.2. PEI-Derived Nanosystems

Cationic polymers are attractive biomaterials for the complexation of nucleic acids, such as siRNAs. Polyethyleneimine (PEI) is one of the most studied, consisting of repeating units of the amine group and two carbon aliphatic CH_2_CH_2_ spacers, which is responsible for high levels of aqueous solubility and also for pH buffering capacity in the endosomal/lysosomal pathway. However, despite allowing efficient siRNA complexation through electrostatic interactions, PEI also exerts toxic effects on cells, depending on its structure and on the cell type tested [59]. To reduce its toxicity, PEI has been conjugated to other polymers such as chitosan, hyaluronic acid (HA), cyclodextrins (CD), and PEG in order to form NPs that are able to protect siRNAs and facilitate endosomal escape.

Chitosan is a biodegradable, biocompatible, and non-toxic polymer, which has a low transfection efficiency for siRNAs [60]. However, this drawback can be mitigated by coupling chitosan to PEI. For example, glycol-chitosan (GC)-PEI-siRNA NPs developed by Huh and colleagues exhibited strong tumor accumulation and target downregulation in vivo [61]. Despite being a promising study, the authors did not go on to provide evidence of in vivo anti-tumoral activity, since the NPs were designed to carry an siRNA targeting red fluorescent protein (RFP), expressed by the xenograft tumors. Zhang et al., developed a dual-targeting chitosan-PEI nanosystem, incorporating the antineoplastic drug Lonidamine. The attached siRNA, targeting the apoptosis inhibitor protein Bcl-2, was covered by a layer of PEG-Poly(acrylic acid)-folic acid to prolong circulation and enhance cancer cell targeting. The mitochondria-targeting ligand triphenylphosphine (TPP) was incorporated into the chitosan-PEI polymer, enhancing the mitochondrial activity of Lonidamine. Although this formulation showed promising results in vitro, the analysis of its in vivo activity remains to be determined [62].

HA is a polysaccharide composed of disaccharide units of d-glucuronic acid, alternating with *N*-acetyl d-glucosamine (NAG), and it is a component of the ECM and synovial fluids. It is anionic, biodegradable, non-toxic, non-immunogenic, and has been used in targeting CD44, which is highly expressed by a variety of tumors [63]. HA-PEI/PEG NPs have been developed by Ganesh and colleagues to carry siRNAs targeting the protein SSB and PLK1, demonstrating in vivo siRNA activity in xenograft tumor models. However, knockdown efficiency was highly dependent on tumor vascularization, suggesting that tumor accumulation was not solely mediated by the active targeting ligand HA [64].

CD are cyclic oligosaccharides, consisting of a macrocyclic ring of glucose subunits joined by α-1,4 glycosidic bonds. β-CD nanosystems are characterized by a hydrophobic interior and hydrophilic exterior and they are used to enhance pharmacokinetics properties of loaded hydrophobic drugs whereas the cationic polyamine backbone allows an electrostatic interaction with siRNAs making them ideal candidates for synergistic drug delivery [65]. Wang and colleagues developed a CD-PEI conjugate adsorbed to gold nanorods for the codelivery of docetaxel (DTX) together with siRNAs targeting the protein p65. The authors showed that the NF-κB pathway and its downstream cascade were inhibited by p65 blockade leading to an enhanced DTX effect and reduced tumor growth in vivo. Interestingly, the nanoplatform could facilitate tunable hyperthermia upon irradiation with a near-infrared (NIR) laser, thus triggering DTX release from the DTX-CD-PEI-siRNA complexes and inducing siRNA endosomal escape. Furthermore, the p65 knockdown sensitized 4T1 breast cells to DTX treatment by suppressing the expression of the anti-apoptotic gene, Bcl-2. The treatment of 4T1 tumor-bearing mice with DTX-CD-PEI-siRNA nanorods inhibited primary tumor growth and reduced the formation of lung metastasis [66]. These results strongly suggest a synergistic effect of NIR irradiation, DTX, and p65 siRNA on tumor cells and the tumor vasculature, demonstrating considerable translational potential.

### 3.3. PLL-Derived Nanosystems

Similar to PEI, poly-L-lysine (PLL) is widely studied for the development of nanocarriers for nucleic acid delivery. PLL has a better biocompatibility and biodegradability profile than PEI [67]. However, PLL/siRNA and PLL-PEG/siRNA polyplexes are more prone to interactions with serum proteins, reducing their ability to knock down target mRNAs [68]. These findings suggest that serum albumin and other polyanions can compete with siRNAs for binding to PLL, leading to particle instability and siRNA disassembly. Interestingly, PLL derivatives are able to mitigate this drawback by enhancing serum stability. For example, polycaprolactone (PCL) has been used to develop a novel PEG-PCL-PLL polymer, which is able to bind siRNAs and form micelles [69]. The assessed in vitro silencing efficiency for this formulation was similar to lipofectamine 2000 and superior to PEI. More recently, Xiao and colleagues developed PEG-PCL-PLL NPs, incorporating platinum-based chemotherapeutic agents (oxaliplatin and cisplatin) and Bcl-2-targeting siRNA for cancer treatment. In vitro experiments showed strong downregulation of Bcl-2 mRNA levels in MCF-7 and OVCAR-4 breast cancer cell lines. In addition, this formulation induced cell death was up to 100-fold more efficient than the free drugs in all the cancer cell lines tested [70]. However, these NPs were not tested in murine tumor models.

PLL has also been conjugated to melanin, a biocompatible pigment, to take advantage of its excellent photothermal properties. Melanin generates heat under NIR irradiation [71], which may be used to facilitate siRNA endosomal escape. The generated melanin-PLL polymer was loaded with *survivin*-targeting siRNA and exhibited a strong inhibitory effect on 4T1 tumor cell growth, both in vitro and in vivo [8]. Furthermore, the development of lung metastases was greatly reduced in treated mice compared to controls. In another report, Sun and colleagues developed a novel triblock polymer composed of poly aspartyl (*N*-(*N*′,*N*′-diisopropylamino ethyl)) (PAD) conjugated to PLL-PEG. PAD was used to confer pH responsiveness to the polymer, thus enhancing the siRNA and DOX delivery to cancer cells. These NPs were loaded with DOX as an antineoplastic agent, and Bcl-2-targeting siRNA to induce apoptosis in cancer cells. Biodistribution analysis in HepG2/adriamycin (ADM) tumor-bearing mice showed that NP accumulation was observed at the tumor site 6 h after injection, and reached a maximum at 24 h, lasting for as long as 48 h. Furthermore, the treatment with DOX/siRNA-loaded PAD-PEG-PLL NPs was able to reduce tumor growth and increase the survival of tumor-bearing mice compared to controls. Lastly, ex-vivo analysis of the tumors showed reduced Bcl-2 and Ki67 expression after treatment, suggesting the induction of apoptosis and reduced cellular proliferation, which led to the control of tumor growth in the treated mice [72]. Another similar nanosystem was recently developed by Wang and colleagues incorporating a disulfide bridge between PEG and PLL (PEG-SS-PLL) to induce PEG release in the endosomes and facilitation of siRNA delivery to the cytoplasm of cancer cells. A VEGF-targeting siRNA was included in the NP to reduce angiogenesis in the TME. In vivo efficacy of the formulation was demonstrated in a HepG2 xenograft murine model. PEG-SS-PLL-siVEGF NP treated mice showed reduced tumor growth compared to controls. Furthermore, histopathology and Western blot analysis of excised tumors showed reduced VEGF expression [73].

### 3.4. Anti-Tumor Nanovaccines Enhanced by siRNAs

Therapeutic cancer vaccines aim to induce de-novo immune responses against cancer cells by promoting the activation and subsequent expansion of tumor-specific CD8^+^ or CD4^+^ T cells, which mediate anti-tumor immunity. NP-based cancer vaccines can help to improve antigen recognition and presentation by APCs (i.e., DCs). The incorporation of antigens in NPs can be achieved by covalent linkage of a protein or a peptide to components of the nanostructure. In addition, nucleic acids such as mRNA and DNA can be attached through electrostatic interactions to the surface of NPs (similarly to siRNAs) and can be processed and translated by APCs into antigenic peptides. Moreover, DNA and mRNA-based cancer vaccines are able to incorporate multiple antigens to increase immunogenicity and the activation of a strong and specific anti-cancer immune response. NP-mediated transfection with DNA or RNA coding for oncogenic proteins or peptides has the advantage of more closely mimicking live infections by incorporating multiple antigen epitopes into one construct. In addition, nucleic acids can serve as self-adjuvants, stimulating the endosomal toll-like receptors (TLR 3, 7, 8, or 9) [74].

Not all NP-based cancer vaccines are required to reach the TME in order to be effective. Tumor targeting of nanovaccines can result in modification of the tumor immune infiltrate, due to the immunomodulatory activity of the adjuvant included in the NP, leading to enhanced anti-tumor responses [75]. Nanovaccines are usually administered subcutaneously, where they form a depot, which causes local inflammation and infiltration of APCs able to take up and process the NP. After activation, DCs then migrate to the lymph nodes, where they present the antigen via the major histocompatibility complex (MHC) class I or II to CD8^+^ or CD4^+^ T cells, respectively. NPs can also be designed to drain directly into the lymphatic system without forming a local depot. For this purpose, NPs ranging from 30 to 100 nm have been shown to effectively reach lymph-nodes after subcutaneous injection while NPs of a larger size are unable to drain effectively into the lymphatic system and are retained at the injection site [14].

The activity of cancer vaccines can be enhanced by the inclusion of siRNAs targeting one or more immune-related proteins aimed at further enhancing function and antigen presentation by APCs. Other siRNA targets include checkpoint blockade inhibitors, which dampen ongoing immune responses in the TME [76]. Thus, including siRNAs in nanovaccines can further enhance the specificity of anti-tumor immunity. Recently, Huang and colleagues designed tumor-targeted lipid dendrimers for hepatocellular carcinoma (HCC) treatment. These NPs consisted of the antigenic molecule hemagglutinin (expressed by the implanted HCC cell line in mice), a PD-L1 siRNA, and an IL-2 expressing plasmid to enhance effector T cell activity. These NPs provide adjuvant activity by inducing the STING pathway, which triggers the secretion of inflammatory cytokines such as CCL5, CXCL10, and IFN-β to further enhance immune cell activity in the TME. In vivo experiments on HCC murine xenografts demonstrated increased tumoral infiltration of CD8^+^ T cells, primary tumor growth suppression, and inhibition of distal metastasis [77]. A PD-L1 siRNA was also introduced in lipid-coated calcium phosphate NPs by Wang and colleagues for the treatment of melanoma [78]. These NPs had intrinsic adjuvant activity on DCs, as they were capable of inducing upregulation of costimulatory molecules, such as CD80 and CD86. They were designed to be transported to the lymph nodes, to induce activation of DCs and co-deliver the melanoma-expressed tyrosinase-related protein 2 (TRP2) peptide in the form of an mRNA vaccine. The anti-tumor efficacy of the siRNA loaded NP containing TRP2 mRNA was tested in B16 melanoma xenografts resulting in a dramatic reduction of tumor growth compared to NPs composed of only one of the two components. Interestingly, the inclusion of a PD-L1-targeting siRNA in the NPs was able to induce a stronger tumor regression compared to a combination of an anti PD-L1 monoclonal antibody (mAb) together with NPs without the siRNA. Furthermore, the co-delivery of siRNA and antigen into the same APCs promoted a significantly higher T cell priming efficiency than that induced by the combination of PD-L1 mAb and mRNA vaccine, suggesting that anti-PD-L1 therapy targeted at the lymphatic system could be superior to systemic administration of anti-PD-L1 mAbs [78]. The efficacy of TRP2 peptides in generating anti-melanoma immune responses was also tested using a novel microparticle composed of yeast-expressing TRP2 peptide-loaded with PEI/siRNA conjugates for the co-delivery of indoleamine 2,3-dioxygenase (IDO1)-targeting siRNAs. IDO1 is expressed in immunosuppressive DCs, and its expression is negatively correlated with survival in melanoma patients [79]. Subcutaneous injection of yeast microparticles containing an IDO1 siRNA was able to significantly delay melanoma tumor growth in vivo, compared to yeast loaded with only TRP2 peptides or IDO1 siRNA, suggesting a strong synergistic effect of the siRNA and TRP2 antigen. In addition, immunohistochemical analysis of tumors revealed a lower amount of T regulatory cells (Tregs) infiltration while CD3^+^ T cell infiltration was enhanced [80].

A microparticle formulation derived from the bacteria, Propionibacterium acnes, called MIS416, was covalently attached to the model antigenic peptide, SIINFEKL, to study its potential as a cancer vaccine. The adjuvant properties of MIS416 were conferred through its cell wall skeleton, consisting of immunostimulatory muramyl dipeptide repeats and CpG sequences which, respectively, activated NOD-2 and TLR-9 receptors to induce DC activation. SIINFEKL was conjugated to MIS416, utilizing a streptavidin bridge between biotinylated versions of MIS416 and SIINFEKL. This formulation was able to enhance costimulatory molecules on treated DCs and induced strong antigen presentation on MHC molecules. Furthermore, in vivo cytotoxicity experiments with the MIS416-SIINFEKL conjugate resulted in the induction of a specific anti-SIINFEKL immune response [81]. In a follow-on study, the feasibility of using MIS416 to deliver signal transducer and activator of transcription *3* (STAT3)-targeting siRNAs was further explored to enhance DC function. A biotinylated STAT3 siRNA, which was conjugated to MIS416 through a disulfide linkage, allowed endosomal escape of the siRNA, and following the treatment of DCs with MIS416-SS-siStat3 the downregulation of both STAT3 mRNA and STAT3 protein levels were observed compared to controls. These studies suggest that an siRNA gene targeting approach could potentially be used to further enhance the cancer vaccine capabilities of MIS416 [82].

## 4. Ongoing Clinical Trials Using siRNA-Loaded NPs for Cancer Therapy

RNAi approaches represent a promising avenue for further investigation in cancer therapy, as they are considered less toxic than classical chemotherapy, and less invasive than surgery. Moreover, they have the potential to be used in conjunction with existing or future treatment modalities to generate a more personalized and efficacious treatment outcome for patients. For example, RNAi NPs could be used in combination with chemotherapy to reduce systemic toxicity, by guiding the delivery of the loaded drug to the TME, as well as further improving treatment efficacy with siRNAs targeting resistance pathways in cancer cells (Table 2).

The first clinical trial regarding NP-mediated siRNA delivery for cancer treatment was published by Davis et al., in 2010 [92]. The NP developed (CALAA-01, produced by Calando Pharmaceuticals) was composed of: (1) a linear cyclodextrin-based polymer (CDP), (2) a human transferrin protein (TF) to engage TF receptors (TFR) on the surface of the cancer cells, (3) external PEG chains to promote nanoparticle stability in biological fluids, and (4) an siRNA targeting the M2 subunit of the ribonucleotide reductase protein (RRM2). Ribonucleotide reductase catalyzes the reduction of ribonucleotides into deoxyribonucleotides. Moreover, inactivation of the ribonucleotide reductase enzyme induces apoptosis in many cancer cell lines [93]. The results of this study also showed convincing intratumoral downregulation of the target protein. However, this study was only performed on a small number of patients, and the results are therefore preliminary.

In 2014, a liposomal RNA interference therapeutic called Atu027, which was produced by Silence Therapeutics GmbH and targeted the protein kinase N3, was evaluated in a dose-escalation phase I clinical trial, resulting in disease stabilization for 41% of patients. The Atu027 structure is composed of the positively charged AtuFect01, a neutral, fusogenic DPhyPE helperlipid and the PEGylated lipid MPEG-2000-DSPE (molar ratio: 50/49/1) [84]. The efficacy of Atu027 is also being tested in a clinical trial together with gemcitabine for the treatment of advanced or metastatic pancreatic cancer [86].

A lipid formulation comprising two different siRNAs targeting both vascular endothelial growth factor (VEGF) and kinesin spindle protein (KSP) (called ALN-VSP) was recently tested on 41 patients with both hepatic and extrahepatic tumors. Almost all patients were heavily pretreated with either chemotherapy and/or anti-VEGF/VEGF receptor (VEGFR) agents. From a safety standpoint, ALN-VSP was generally well-tolerated and compared favorably to chemotherapy and to other orally or intravenously administered targeted therapies in oncology. The results were encouraging with one patient achieving a complete response, while other patients had stable disease at all sites for approximately 8–12 months. In addition, tumor biopsies showed effective siRNA targeting and downregulation of target proteins [94].

Another lipid formulation called EnCore lipid nanoparticles (DCR-PHXC-101) was developed by Dicerna pharmaceuticals to downregulate the expression of the transcription factor Myc, which is dysregulated in many human tumors. A phase I dose-escalation study evaluated the safety, pharmacokinetics, pharmacodynamics, and clinical activity of DCR-MYC in patients with advanced solid tumors, multiple myeloma or lymphoma. One patient experienced a complete metabolic response, which was sustained for >8 months without further treatment. Evidence of metabolic responses after cycle 1 and tumor shrinkage were observed in multiple patients [95]. An update on the results of this study is expected soon [89].

A DOPC-based liposomal formulation targeting the tyrosine kinase EphA2 is currently under investigation in a phase I clinical trial in patients with advanced and recurring solid tumors [96]. The results from this are expected to be published at the end of 2020. EphA2 is overexpressed in many patients and is associated with a poor prognosis and immunosuppression [97]. Pre-clinical trials in mice and *Rhesus* macaques were encouraging, and no gross pathological or dose-related microscopic findings were observed in either the acute (24 h), or recovery (14 and 28 days) phases [96].

Lastly, although not NP-based, an interesting therapeutic platform for local and prolonged delivery of siRNA was developed by Silenseed Ltd. for intra-tumoral treatment. This device, loaded with *KRASG12D*-targeting siRNA, is based on encompassing the siRNA with a miniature biodegradable polymeric matrix that protects and enables siRNA release for an extended period of time, regionally within tumor tissue. A phase I/IIa clinical study was initiated in patients with non-operable locally advanced pancreatic cancer in combination with gemtabicine. Results were encouraging with 12 of 15 patients showing no evidence of tumor progression and the majority (10 of 12) demonstrating stable disease with two partial responses [98]. A multinational randomized phase 2b clinical trial is currently in progress [91].

Regarding the use of cancer vaccines in clinical trials, many studies have utilized engineered autologous DCs in combination with siRNAs, which are re-infused in patients to induce a de-novo immune response against tumor antigens. For example, Wand and colleagues tested the therapeutic efficacy of a *SOCS1*-silenced DC vaccine loaded with two tumor-associated antigens, survivin and MUC-1 in patients with relapsed acute leukemia (AL) after allogenic hematopoietic stem cell transplantation (allo-HSCT). However, to our knowledge, NP-based cancer vaccines have yet to reach the clinical stages of development. Although tumor-specific T cell responses have been observed in preclinical studies, the implementation of cancer vaccines in clinical trials has failed to achieve good responses, mainly due to the ongoing immune suppression in the TME [99]. However, the recent development of immune checkpoint blockade therapeutics could dramatically enhance the potential of cancer vaccines as previously described [76].

## 5. Concluding Remarks

The potent gene silencing capacity of siRNAs is a feature that bodes for their use as ideal novel cancer therapeutics. The complexation of siRNAs to bio-compatible cationic polymers has paved the way for the development of a large array of delivery systems, able to effectively induce specific gene silencing in target cells. These nanosystems allow the co-delivery of siRNAs together with chemotherapeutics to take advantage of synergistic anti-cancer effects in vivo. In addition, nanosystems are highly customizable and can be targeted to cancer cells by adding targeting ligands to their surface. By tailoring NPs to be patient-specific, a higher dose of both siRNAs and chemotherapeutics could reach the TME, thus limiting systemic toxicity and improving efficacy. In addition, siRNAs are incorporated into a delivery system by electrostatic interactions with its cationic components, which confers high siRNA protection, high loading efficiency and facilitates siRNAs release into the cytoplasm of target cells, making these systems ideal. Both PEI and PLL-based nanosystems are highly customizable and compete in terms of loading capacity and delivery efficacy with cationic phospholipids such as DOPE and DOTAP. Encouraging results obtained in pre-clinical studies have not always translated to good outcomes in human patients, and very few formulations have to date reached phase II clinical trials. However, the potential for combination with immune checkpoint blockade therapy could further enhance the therapeutic potential of siRNA-loaded NPs, especially in the case of cancer nanovaccines. Recent discoveries are propelling us beyond the physiological and intracellular barriers that limit siRNA functionality, and we predict, that in due course the remarkable translational potential of RNAi will eventually find its place in the oncology clinic.

## Figures and Tables

**Figure 1 molecules-25-02692-f001:**
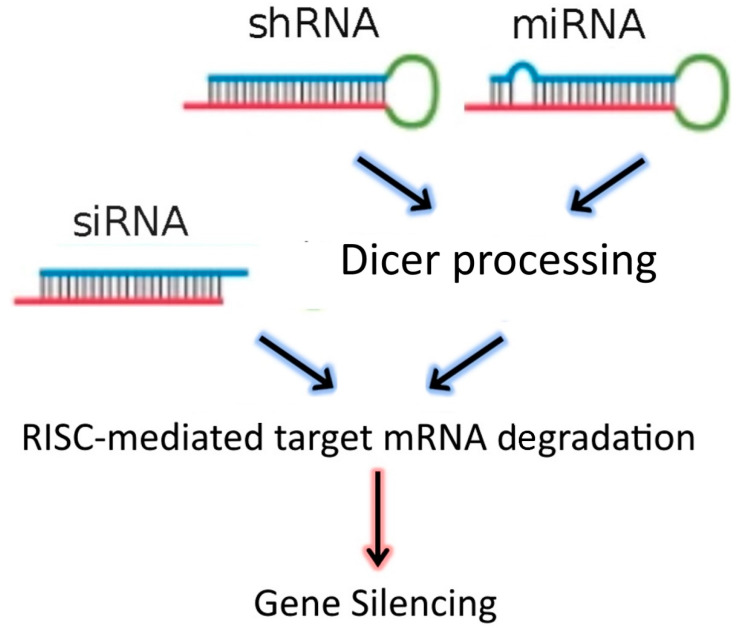
RNAi-based therapeutics for gene silencing.

**Figure 2 molecules-25-02692-f002:**
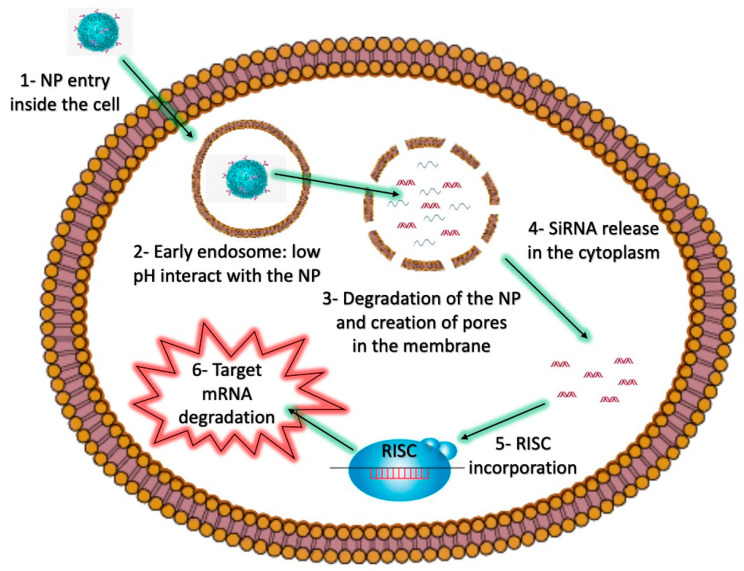
The intracellular barriers of siRNA-loaded NPs as nanovectors.

**Table 1 molecules-25-02692-t001:** Physiological barriers in siRNA delivery by intravenous injection.

Barrier	Approach
Degradation by RNAses	Chemical modification of siRNAs, inclusion of siRNAs in NP-based delivery systems
Renal clearance	Inclusion of the siRNA in a nanocomplex with a HD >6 nm
Reticuloendothelial system	Addition of PEG to the nanocomplex to reduce protein corona formation and phagocytosis
Limited access into tumor tissue	Passive accumulation: limit NP size (<200 nm) to promote the EPR effect. Active targeting: Inclusion of a targeting ligand on the surface of the NPs

**Table 2 molecules-25-02692-t002:** Clinical trials with SiRNA-loaded vectors.

Name	Type	Target	Type of Cancer	Status	Reference
CALAA-01	Cyclodextrin polymer-based NPs	RRM2	Solid tumors	Completed	NCT00689065 [83]
Atu027	Liposomes	Protein kinase N3	Solid tumors, pancreatic carcinoma	Completed Completed Completed	NCT00938574 [84] NCT00938574 [85] NCT01808638 [86]
ALN-VSP	Lipid-based NPs	VEGF and KSP	Solid tumors	Completed Completed	NCT00882180 [87] NCT01158079 [88]
DCR-PHXC-101	Lipid-based NPs	Myc	Solid tumors, multiple myeloma, non-Hodgkin’s lymphoma	Terminated	NCT02110563 [89]
SiRNA-EphA2	Liposomes	EphA2	Advanced cancers	Recruiting	NCT01591356 [90]
siG12D LODER	Biodegradable polymeric matrix	KRASG12D	Pancreatic ductal adenocarcinoma, pancreatic cancer	Recruiting	NCT01676259 [91]

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
