# Peer review of "Lipid and Polymer-Based Nanoparticle siRNA Delivery Systems for Cancer Therapy"

_molecules, 2020, doi:10.3390/molecules25112692_

Round 1

Reviewer 1 Report

The authors presented an interesting issue concerning siRNA delivery in lipid nanocarriers. The review is well written and designed. Some comments should be addressed:

  • in the final paragraph "conclusions" should be highlighted the most beneficial system for siRNA encapsulation and which characterizes by the highest delivery efficacy
  • physical delivery methods should be included e.g. electroporation or sonoporation, these methods are more efficient than chemical methods and often can improve the efficacy in case not optimized surface charge of nanocarriers. 

Author Response

We would like to thank the reviewer for their comments. We have revised the manuscript taking into account the reviewer's comments.

We provide a point-by-point response to the reviewer’s comments copied here in italics, and our responses follow in non-italics.

1) in the final paragraph "conclusions" should be highlighted the most beneficial system for siRNA encapsulation and which characterizes by the highest delivery efficacy

Response: We added a sentence in the conclusions (page 13, lines 539-541) to answer the reviewer’s comment, which reads as follows; “Both PEI and PLL-based nanosystems are highly customizable and compete in terms of loading capacity and delivery efficacy with cationic phospholipids such as DOPE and DOTAP.”

However, it is hard to identify the ‘’best’’ delivery system in general, since experiments carried out by different groups were not always comparable. We have identified both PEI and PLL-based nanosystems as ideal candidates together with lipoplexes since these systems ensure high siRNA loading and high delivery efficacy in general.

2) physical delivery methods should be included e.g. electroporation or sonoporation, these methods are more efficient than chemical methods and often can improve the efficacy in case not optimized surface charge of nanocarriers. 

Response: We have added several new sentences at the start of section 2.1 (page 4, lines 117 to 123), as follows; “Methodologies such as electroporation and sonoporation have been used to induce the transfer of genetic material inside cells by creating transient pores on the cell membranes. However, these delivery techniques have limitations for in vivo delivery, such as lack of tissue penetration, cell and tissue damage induction and frequently very low transfection efficiency. Therefore, more discreet and efficient methods are desirable to achieve siRNA delivery to tumor cells. Systemic delivery via intravenous administration is the strategy of choice when the target sites are not locally confined or not readily accessible, such as the vast majority of tumors in humans.”

Regarding in vivo delivery of siRNA for cancer treatment, electroporation and sonoporation lack sufficient tissue penetration capacity to have a significant impact on most tumors in pre-clinical or clinical studies.

Reviewer 2 Report

Review for molecules-818521 “Lipid and polymer-based nanoparticle siRNA delivery systems for cancer therapy: an update”

The manuscript title promises an update. Therefore, the reader expects a summary of new trends, technologies, advances in efficacy, and translation.

Minor update compared to reviews like (doi: 10.2147/IJN.S200253; doi: 10.1016/j.trsl.2019.07.006 both from the year 2019) – which are more complete for delivery strategies. To be useful a systematic analysis of fields with advances/break through technologies, best successes in translation, specialization on specific field(s) with current updates are preferred to listing of many examples.  

The manuscript could state more clearly that it focuses its review on delivery via the blood (no discussion on options of local therapy). In particular, the extracellular barrier section would profit from a clear statement – since barriers for local delivery could be very different.

Figure 1 in the current form is not providing much information. Including here miRNA, which is not the topic of this review, is somewhat distracting. The more important message to put in the pharmaceutical technology world would be the difference between Dicer-dependent and Dicer-independent RNAi – which was an important finding to advance RNAi as a therapeutic strategy.

Table 1 should be improved: Caption title should include that this are “barriers” to i.v. delivery. The table lists rather challenges in the development of siRNA delivery systems (e.g., “accumulation in the TME” is no barrier; either it is a challenge to achieve specific accumulation – or the barrier is a high interstitial pressure and limited access into tumor tissue). Further, corona formation and RES clearance of particles are interrelated – since opsonization is a trigger for phagocytotic removal. PEGylation is the most often used but not only way to reduce corona formation (write e.g., addition of PEG).

EPR effect is not listed in Tab. 1. For i.v. delivered carriers the presence of EPR effect in some tumor types facilitates the accumulation of particles of the correct size – without targeting specific ligands. Other tumor types show less or no EPR effect. Thus, the variable amount of EPR could be included in the challenges/barriers. Among others, studies of T. Lammers/ F. Kiessling team show the importance of permeability and chances to enhance permeability.

“In addition, active targeting reduces off-target effects of NPs by limiting their effect in non-target cells.” This statement simplifies the effect of targeting. Active targeting can enhance the uptake of cells with the respective addressed receptor to a certain percentage when reaching the target tissue. Active targeting per see does not guide the particle. Selectivity for tumor cells can also be achieved by siRNA target gene selection (more than probably possible with typical chemotherapy).

Figure 2 suggests that all nanocarriers for siRNA delivery have similar behavior in endosomal escape. Valid for liposomal and polymeric carriers?

The issue of siRNA as stimulants of innate immune responses via toll-like receptors can be quite well controlled (e.g, correct length, nucleotide modifications, algorithms to predict good siRNA sequences) (for a review see: https://doi.org/10.1146/annurev-chembioeng-061010-114133). The short mentioning in the “challenges” is somewhat unclear. In particular, as it can be even wanted in tumor therapy to have immune stimulation.

An explanation for the focus on lipid and polymeric systems is missing. Why did the authors decide to omit from this review other carrier types, e.g. attempts of combined systems using polymers + lipids, more innovative polymers (maybe also coming from other nucleotide carriers or non-cancer approaches– but adaptable to siRNA delivery for tumor therapy) or the advanced in siRNA conjugate delivery? Any new relevant patents in delivery technologies for siRNA?

The nanovaccine section could be shorter (omitting some detailed description of the general concept of vaccination in tumor therapy) but highlighting the potential of combination with siRNA by explaining more clear what target genes for siRNA are suitable to assist synergistically. Why a co-delivery is advantageous?

Table 2 on clinical trials should be improved by including a “current state” column (to show which trials are not propagated any more (e.g., CALAA-01) and which are still ongoing or even new. Clinical trial numbers would be preferred for paper references.

Doublets in the references: References 93 and 94 are identical. Reference 90 and 95 are identical.

Author Response

We would like to thank the reviewer for their comments. We have extensively revised the manuscript taking into account the reviewer's comments.

We provide a point-by-point response to each of the reviewer’s comments in the following; the reviewer's comments are in italics, and our responses follow in non-italics.

1) The manuscript title promises an update. Therefore, the reader expects a summary of new trends, technologies, advances in efficacy, and translation.

Minor update compared to reviews like (doi: 10.2147/IJN.S200253; doi: 10.1016/j.trsl.2019.07.006 both from the year 2019) – which are more complete for delivery strategies. To be useful a systematic analysis of fields with advances/break through technologies, best successes in translation, specialization on specific field(s) with current updates are preferred to listing of many examples.  

Response: Our review is not primarily focused on providing a description of various delivery strategies for siRNAs (which are described more in depth in the reviews suggested). Our review focuses on delivery systems based on biocompatible cationic polymers and lipids utilized for the delivery of siRNAs in combination with chemotherapy (for direct tumor targeting) or with adjuvants and peptides (for cancer vaccines). We decided to provide a number of examples rather than describe advances in general because this review is aimed at describing in detail the delivery systems used (in the last few years) rather than describing a current update on the field in general. However, we agree that the original title might suggest a different type of review to certain readers, therefore we have removed the words “:an update” from the revised title.

2) The manuscript could state more clearly that it focuses its review on delivery via the blood (no discussion on options of local therapy). In particular, the extracellular barrier section would profit from a clear statement – since barriers for local delivery could be very different.

Response: Two new parts were added as suggested by the reviewer, as follows:

  1. i) At the end of the Introduction (page 3, line 96-97), the sentence “This review discusses primarily the delivery of NPs by the intravenous route.” Has been added to say that the review is primarily focused on the delivery of NPs by the intravenous route.
  2. ii) In the section regarding the extracellular barriers (page 4, lines 121-123) the sentence “Systemic delivery via intravenous administration is the strategy of choice when the target sites are not locally confined or not readily accessible, such as in the vast majority of tumors in humans.” Has been added to highlight that the intravenous route is the strategy of choice when the target sites are not locally confined or not readily accessible, such as the vast majority of tumors

3) Figure 1 in the current form is not providing much information. Including here miRNA, which is not the topic of this review, is somewhat distracting. The more important message to put in the pharmaceutical technology world would be the difference between Dicer-dependent and Dicer-independent RNAi – which was an important finding to advance RNAi as a therapeutic strategy.

Response: Figure 1 has been modified to include the distinction between Dicer-dependent and Dicer-independent RNAi processing, in which this difference is highlighted between siRNAs and shRNAs/miRNAs. The rationale for including a brief mention of shRNAs is because shRNAs are frequently used as an alternative to siRNAs, although shRNAs require Dicer-dependent processing, like miRNAs.

4) Table 1 should be improved: Caption title should include that this are “barriers” to i.v. delivery. The table lists rather challenges in the development of siRNA delivery systems (e.g., “accumulation in the TME” is no barrier; either it is a challenge to achieve specific accumulation – or the barrier is a high interstitial pressure and limited access into tumor tissue). Further, corona formation and RES clearance of particles are interrelated – since opsonization is a trigger for phagocytotic removal. PEGylation is the most often used but not only way to reduce corona formation (write e.g., addition of PEG).

EPR effect is not listed in Tab. 1. For i.v. delivered carriers the presence of EPR effect in some tumor types facilitates the accumulation of particles of the correct size – without targeting specific ligands. Other tumor types show less or no EPR effect. Thus, the variable amount of EPR could be included in the challenges/barriers. Among others, studies of T. Lammers/ F. Kiessling team show the importance of permeability and chances to enhance permeability.

Response: Table 1 has been modified according to the reviewer’s suggestions

5) “In addition, active targeting reduces off-target effects of NPs by limiting their effect in non-target cells.” This statement simplifies the effect of targeting. Active targeting can enhance the uptake of cells with the respective addressed receptor to a certain percentage when reaching the target tissue. Active targeting per see does not guide the particle. Selectivity for tumor cells can also be achieved by siRNA target gene selection (more than probably possible with typical chemotherapy).

Response: A new sentence was added to the end of section 2.1 (page 5, lines 166-167), which reads as follows, “To reduce off-target effects, siRNAs can be designed to target cancer-specific mRNAs, thus limiting their effect on non-target cell”. In addition, the sentence on lines 162-164 has been altered to address the reviewer’s comments.

6) Figure 2 suggests that all nanocarriers for siRNA delivery have similar behavior in endosomal escape. Valid for liposomal and polymeric carriers?

Response: Figure 2 aims to describe the intracellular barriers of nanoparticle-associated siRNA delivery and summarizes the general mechanism of action of NPs which results in the release of the loaded siRNA in the cytoplasm. The title in the figure has been altered to reflect this; “Figure 2: The intracellular barriers of siRNA-loaded NPs as nanovectors”. This figure is not focused on the endosomal escape strategies per se. As the reviewer suggested, the strategies utilized to induce siRNA release can be quite different depending on the delivery system used. These strategies are described in the text in section 2.2 (unchanged from the original version).

7). The issue of siRNA as stimulants of innate immune responses via toll-like receptors can be quite well controlled (e.g, correct length, nucleotide modifications, algorithms to predict good siRNA sequences) (for a review see: https://doi.org/10.1146/annurev-chembioeng-061010-114133). The short mentioning in the “challenges” is somewhat unclear. In particular, as it can be even wanted in tumor therapy to have immune stimulation.

Response: A new part was added near the end of section 2.2 (page 6, lines 220-224) to address the reviewer’s comments, which reads as follows, “Both the sequence and structure of siRNAs can be adjusted to minimize the activation of innate immune cells [49]. On the other hand, in the TME immune stimulation locally, due to the activation of immune cells by siRNAs could be beneficial to further enhance the immune response against cancer cells. Therefore, it is important that siRNA characteristics are tailored carefully, so as to either avoid or to induce immune activation”. We included the review citation, as was suggested by the reviewer, and we point out that, as suggested, siRNA characteristics should be chosen carefully to take in consideration the unspecific effect that siRNAs could have on immune cells. We agree that in some cases, local immune stimulation in the TME could be beneficial.

8). An explanation for the focus on lipid and polymeric systems is missing. Why did the authors decide to omit from this review other carrier types, e.g. attempts of combined systems using polymers + lipids, more innovative polymers (maybe also coming from other nucleotide carriers or non-cancer approaches– but adaptable to siRNA delivery for tumor therapy) or the advanced in siRNA conjugate delivery? Any new relevant patents in delivery technologies for siRNA?

Response: We decided to focus the review on describing delivery systems with similar characteristics rather than skimming between a wide variety of different nanosystems. This allowed us to describe in detail the differences between relatively similar systems and to highlight the specific characteristics of individual nanosystems without being too general in the description.

In addition, we decided to focus on delivery systems based on the use of biocompatible polymer and lipids that have been tested adequately. In this regard, many groups around the world have been investigating new ways to produce stable nanoparticles with these materials and how to optimize the formulation composition for siRNA loading. We also described in detail some of the characteristics which are similar to these materials (the ability to complex siRNA forming nanoparticles, and endosomal escape capacity) which we believe makes these material ideal candidates for in vivo siRNA delivery. Near the end of the Introduction (page 3, lines 141-144), we have added the following sentence to help explain the focus of the review; “Moreover, several of the characteristics of these lipid, polymer and inorganic-based NPs, including abilities for the formation of siRNA nanoparticles, and endosomal escape, are characteristics that we believe make these materials ideal candidates for in vivo siRNA delivery.”

The reviewer suggested that we should have included a number of other different and more innovative materials and to outline new patents regarding the potential use of these novel biomaterials for siRNA delivery. We decided that we would only review lipid and polymeric delivery systems for use in siRNA delivery, and given that other materials that have been tested for the delivery of other nucleic acids may not be suitable to deliver siRNAs, as it is not clear whether they would also work well for siRNAs, we therefore didn’t include them. We also decided not to include shRNA and miRNA in detail to avoid confusion and to focus on siRNA delivery only.

9). The nanovaccine section could be shorter (omitting some detailed description of the general concept of vaccination in tumor therapy) but highlighting the potential of combination with siRNA by explaining more clear what target genes for siRNA are suitable to assist synergistically. Why a co-delivery is advantageous?

Response: Following the reviewer’s suggestions, we removed some material from section 3.4 to reduce its length. We have particularly focused this part on the synergistic use of siRNAs targeted to immune checkpoint inhibitors, because many research findings have shown that immune activation synergizes with immune checkpoint blockade therapy, and that blockade of immunosuppressive proteins can also be achieved by siRNAs. We have also highlighted why the codelivery of siRNAs is advantageous citing two examples taken from the literature where the nanosystems tested were compared to siRNA-lacking controls showing a dramatic of reduction in efficacy when the siRNA was not present. In addition, we have also described how anti-PD-L1 therapy targeted to the lymphatic system (with a nanosystem loaded with anti-PD-L1-siRNA) could be superior to systemic administration of anti-PD-L1 mAbs showing the strong synergy effect of the siRNA in this formulation.

Target genes, which were mentioned regarding each formulation, have been discussed in this section.

10) Table 2 on clinical trials should be improved by including a “current state” column (to show which trials are not propagated any more (e.g., CALAA-01) and which are still ongoing or even new. Clinical trial numbers would be preferred for paper references.

Response: Table 2 has been modified according to the reviewer’s suggestions

11) Doublets in the references: References 93 and 94 are identical. Reference 90 and 95 are identical.

Response: References have been corrected

Round 2

Reviewer 2 Report

The authors thoroughly revised the manuscript clearly improving the quality.

The decision to limit to selected very well known excipients, is explained with in depth review of the more narrow field. Such argumentation in general is valid and the review can be published in the current form.

However, a somewhat broader perspective including further biopolymers, more complex lipid formulations and polymer/lipid combinations would have possibly attracted a broader readership.